# Bioactive Nanofiber-Based Conduits in a Peripheral Nerve Gap Management—An Animal Model Study

**DOI:** 10.3390/ijms22115588

**Published:** 2021-05-25

**Authors:** Tomasz Dębski, Ewa Kijeńska-Gawrońska, Aleksandra Zołocińska, Katarzyna Siennicka, Anna Słysz, Wiktor Paskal, Paweł K. Włodarski, Wojciech Święszkowski, Zygmunt Pojda

**Affiliations:** 1Department of Regenerative Medicine, Maria Sklodowska-Curie National Research Institute of Oncology, Wawelska 15B, 02-034 Warsaw, Poland; Aleksandra.Zolocinska@pib-nio.pl (A.Z.); siennicka.katarzyna@wp.pl (K.S.); anna.slysz@pib-nio.pl (A.S.); zygmunt.pojda@pib-nio.pl (Z.P.); 2Centre for Advanced Materials and Technologies CEZAMAT, Warsaw University of Technology, Poleczki 19, 02-822 Warsaw, Poland; ewakijenska@gmail.com; 3Materials Design Division, Faculty of Materials Science and Engineering, Warsaw University of Technology, Woloska 141, 02-507 Warsaw, Poland; wojciech.swieszkowski@pw.edu.pl; 4Centre for Preclinical Research, Department of Methodology, Medical University of Warsaw, Banacha 1b, 02-097 Warsaw, Poland; wiktor.paskal@gmail.com (W.P.); pawel.wlodarski@wum.edu.pl (P.K.W.)

**Keywords:** nerve reconstruction, conduit, ASCs, P(LLA-CL)-COL-PANI, peripheral nerve injury, nerve gap

## Abstract

The aim was to examine the efficiency of a scaffold made of poly (L-lactic acid)-co-poly(ϵ-caprolactone), collagen (COL), polyaniline (PANI), and enriched with adipose-derived stem cells (ASCs) as a nerve conduit in a rat model. P(LLA-CL)-COL-PANI scaffold was optimized and electrospun into a tubular-shaped structure. Adipose tissue from 10 Lewis rats was harvested for ASCs culture. A total of 28 inbred male Lewis rats underwent sciatic nerve transection and excision of a 10 mm nerve trunk fragment. In Group A, the nerve gap remained untouched; in Group B, an excised trunk was used as an autograft; in Group C, nerve stumps were secured with P(LLA-CL)-COL-PANI conduit; in Group D, P(LLA-CL)-COL-PANI conduit was enriched with ASCs. After 6 months of observation, rats were sacrificed. Gastrocnemius muscles and sciatic nerves were harvested for weight, histology analysis, and nerve fiber count analyses. Group A showed advanced atrophy of the muscle, and each intervention (B, C, D) prevented muscle mass decrease (*p* < 0.0001); however, ASCs addition decreased efficiency vs. autograft (*p* < 0.05). Nerve fiber count revealed a superior effect in the nerve fiber density observed in the groups with the use of conduit (D vs. B *p* < 0.0001, C vs. B *p* < 0.001). P(LLA-CL)-COL-PANI conduits with ASCs showed promising results in managing nerve gap by decreasing muscle atrophy.

## 1. Introduction

Peripheral nerve injuries (PNI) are one of the most disabling components of traumatic injuries of extremities, especially traumatic hand injuries, and they may accompany 2.8–6.1% of cases [1,2,3]. Sequels of PNI concern sensory or motor malfunction of an injured limb, causing a severe decrease in quality of life.

In the case of nerve shortening after trauma, neuroma resection, or during reconstructive procedure, a surgeon may encounter a need for management of a gap between nerve stumps. The main goal of the procedure is to provide a tension-less junction between the ends of an injured nerve, since that factor mainly contributes to delayed neuroregeneration [4]. To date, there are few possible surgical approaches in managing peripheral nerve gaps. Namely, primary microsurgical suturing, the use of sutureless techniques (tissue glues), allografts, autografts, and synthetic conduits [5]. However, primary suturing remains the gold standard only in case of tension-free conditions. If the primary approach is impossible, there is a need for creating a bridge between nerve stumps. In the upper extremity, the digital nerve’s gap should be treated with processed nerve allograft (PNA), unless the distance exceeds 3 cm; then, an autograft should be used. Motor or mixed nerves injury of upper extremity always requires the use of an autograft [6]. Autografts require the sacrifice of a donor site of a patient and cause loss of innervation of a certain area, require additional cuts, and there are a limited number of available sites with acceptable morbidity. Processed allografts do not cause donor site morbidity of a patient, but they may not be available in numerous countries (legal, ethical, or organizational issues) and may be used in injuries ranging from 5 to 50 mm [7,8]. Conduits emerged as an alternative to these solutions and struggle their way to the algorithms in the clinical approach. Currently, conduits are successfully used in digital nerve repair and are an effective alternative to PNA in managing short neural gaps [6]. There are a few products approved by the FDA, which are manufactured from type I collagen, polyglycolic acid, polycaprolactone, polyvinyl alcohol, porcine small intestine submucosa [9], or chitosan [10]. Since they are off-the-shelf products, easily stored, accessible, and biocompatible, they pose a serious alternative and direction toward the future solution of nerve repairs.

Thus, exploration of novel material continues, and recently, poly(L-lactic acid co-ϵ-caprolactone)-based scaffolds are perceived as potential candidates in that application [9].

Poly(L-lactic acid co-ϵ-caprolactone) [P(LLA-CL)] is a non-toxic biodegradable polyester possessing good mechanical properties and degradation rates and has been successfully utilized in nerve tissue engineering applications [11]. However, its surface lacks binding sites recognizable for the cells and is highly hydrophobic. Moreover, there are several reports indicating that electrospun polycaprolactone shells of conduits may cause massive inflammatory reaction [12,13]. To overcome these limitations, this synthetic polymer is mixed with natural bio-active agents such as proteins, growth factors, etc. A collagen (COL) compound was chosen for the fabrication of our conduit after a prior optimization study, which revealed superior proneurogenic feature of P(LLA-CL) with collagen compared to pristine synthetic polymer [14]. Polyanilin (PANI) was added because of its electroconductive features, which promote neuroregeneration [15].

Conversely, regenerative medicine techniques are intensively explored in the field of neuroregeneration. Harnessing multipotent cells (adipose-derived stem cells—ASCs, bone marrow stromal cells—BMSCs, mesenchymal stem cells—MSCs) for promoting the regeneration of an injured nerve has been widely reported. Thanks to their pleiotropic effect on inflammation, immunomodulation, neurotrophic factors excretion, and transdifferentiation into Schwann-like cells, they may serve as a precious component of neuroregenerative therapies [16,17]. ASCs are eagerly investigated in nerve regeneration studies due to their easy harvest and confirmed beneficial influence via exosomes promoting Schwann cells proliferation, migration, and myelination [18]. Yet alone, they are an insufficient weapon in PNI with a long nerve gap. Hundepool et al., in their metanalysis, summarized the supportive role of stem cells in nerve conduits [3].

The combination of conduits and stem cells in nerve gap management has been studied in both preclinical and clinical phases [19,20,21,22]. Grimoldi et al. showed promising results in the management of large peripheral nerve lesions with collagen tubes filled with skin-derived autologous stem cells not only in preclinical experiments in rats but also in clinical case reports. In their study, patients demonstrated ongoing recovery of motor and sensory functions of the median nerve during the 3-year follow-up [19].

The aim of the study was to create a novel type of fibrous conduits based on poly(L-lactic acid co-ϵ-caprolactone)/collagen/PANI and investigate their efficacy in the regeneration of a 10 mm gap in the sciatic nerve of a rat with and without ASCs enrichment. To the best of our knowledge, this is the first report in the literature on an electrospun conduit of this composition combined with adipose stem cells in order to promote the regeneration of nerve sciatic nerve.

## 2. Results

### 2.1. Morphology of the Electrospun Tubes

The electrospun tubes had uniform thickness of the wall of 272 ± 12 µm, and the average outer diameter of the tubes was 2.04 ± 0.04 mm. The SEM images of the inner surface of the conduits displayed randomly oriented fibers with relatively uniform morphology and an average diameter of 460 ± 143 nm. (Figure 1 and Table 1).

Wettability measurements revealed the hydrophilic nature of the surface of the conduits, and the average value of the water contact angle measurement for the P(LLA-CL)-COL-PANI was 81.23 ± 2.14°.

Mechanical testing results are shown in Table 1. The ultimate tensile strength obtained for electrospun fibrous conduits was 8.53 ± 1.43 MPa, and the elongation at break was 332.25 ± 72.11%.

### 2.2. Biocompatibility Studies

Metabolic activity (measured as absorbance—Abs) detected for cells cultured on P(LLA-CL)-COL-PANI electrospun fibrous meshes was higher than detected for cells cultured on P(LLA-CL)-PANI, as it is shown in Figure 2A. Moreover, the expression (measured as relative quantification—RQ) of the Neurod 1 gene, which is one of the most important transcription factors in neuronal differentiation, was seven times higher for cells undergoing differentiation on P(LLA-CL)-COL-PANI fibrous meshes (55.70 ± 1.91) than for cells differentiated on P(LLA-CL)-PANI (8.23 ± 0.56) and almost two times higher for cells differentiated on plastic plates (29.95 ± 0.3), which can be seen in Figure 2B.

During in vitro testing of the tubular P(LLA-CL)-COL-PANI conduits, it was observed that ASCs were moving out from the tubes and were present within the culture Petri dishes; however, they were present in a lower number than inside the tube 0.217 × 10^6^ ± 0.017 cells vs. 0.165 × 10^6^ ± 0.015 cells and with lower viability 85% ± 8 vs. 74.8% ± 8.

### 2.3. Animal Study and SFI

Observation of operated animals lasted 6 months. One rat from group B was euthanatized earlier due to diagnosed actinomyces infection (21st week of observation). No wound dehiscence or autotomies were observed. In monthly intervals, SFI was calculated from gait analysis. There was no statistically significant difference in SFI mean values among experimental groups (B, C, D). However, group D achieved the highest results over the observation period (Figure 3). At the time of rats’ euthanasia and samples harvest, no excessive scarring, fibrosis, or adhesions were observed in any group (Figure 4).

### 2.4. Muscular Tissue

Gastrocnemius muscles were excised in both hind limbs and weighted. Macroscopically, muscles from the non-operated side were bigger vs. ipsilateral to SN lesion (Figure 5). Microscopic images of GM muscles are presented in Figure 6. It was confirmed with weight analysis, which revealed a maximal ratio 0.82 in Group B; also, the mean muscle mass ratio was highest among examined groups (mean ± SD; 0.77 ± 0.05). An insignificantly lower mean muscle mass ratio was observed in Group C (0.74 ± 0.04). Group D showed the lowest mean muscle mass ratio among Groups B, C, and D (0.67 ± 0.07), but statistical significance was observed only when compared to Group B (*p* < 0.05). Each group showed significantly higher mean muscle mass ratio vs. Group A (*p* < 0.0001) (Figure 7). Analyses of fibrosis of muscle samples are presented in Figure 8 and Table 2. Group A had significantly the highest level of fibrosis (65.34 ± 12.3) versus any other group (*p* < 0.001), whereas the group of healthy muscle samples showed the lowest percentage of fibrotic tissue (5.11 ± 2.23, *p* < 0.001). Experimental groups B, C, and D showed respectively 13.61% ± 3.42, 15.38% ± 2.81, and 12.11% ± 2.76 of fibrotic tissue in a section, without significant differences among groups (*p* > 0.05).

### 2.5. Neural Tissue

Along with the muscular tissue, fragments of SN were harvested in the final timepoint. NF-200 staining reveals nerve fibers and enables their identification in histological samples. As presented in Figure 4, in each group with nerve reconstruction nerve continuity was achieved. In Group B, autografts presented less compacted appearance vs. Groups C and D. In Group D, perineurium were less hyperplastic vs. Group C (Figure 9). In quantitative analysis, Group D showed the highest nerve fiber density (mean ± SD; 1152 ± 59.16) compared to Groups C (1034 ± 67.76) and B (803.6 ± 43.47). The observation was statistically significant (*p* < 0.001 and *p* < 0.0001 respectively) (Figure 10A). Further analyses of nerve fibers morphometry are presented in Figure 10 obtained with WP staining (Figure 11). Axon density revealed consistent results with NF-200 staining (highest density in Group D, then in C and lowest in Group B). Axonal area was lowest in Group C, and it was significantly lower in B and D (*p* = 0.0001; *p* = 0.003). Myelin thickness was highest in Group B, and it was significantly higher than Group C (0.25 ± 0.01 vs. 0.19 ± 0.03, *p* = 0.0002) and insignificantly higher vs. Group D (0.25 ± 0.01 vs. 0.22 ± 0.02, *p* = 0.22). The circularity ratio was highest in Group D (*p* = 0.03 and *p* = 0.003), followed by C, and it was the lowest in the autograft group (respectively 0.71 ± 0.03; 0.66 ± 0.03; 0.64 ± 0.04). The G ratio revealed significantly higher mean values in Groups D (0.42 ± 0.04) vs. B (0.33 ± 0.04) and C (0.26 ±0.03) (*p* = 0.0007 and *p* = 0.0001) and lower in C vs. B (*p* = 0.006). No neuroma was found on nerve cross-sections.

## 3. Discussion

In this study, a novel P(LLA-CL)-COL-PANI scaffold, with or without ASC enrichment, was analyzed in the context of managing peripheral nerve gaps in a rat model. We aimed to confirm the biocompatibility of the new conduit and compare its efficiency versus the gold standard—a nerve autograft.

Electrospinning has been recently found to be a successful method for fibrous nerve conduits’ fabrication. Wang et al. in their study developed a polyblend electrospun conduit based on a polyblend of P(LLA-CL) and silk fibroin. However, although the conduits possessed good biocompatibility and showed improvement of tissue regeneration compared to the non-treated injured site, their in vivo result using a rat model presents still low outcome compared to nerve autograft [23]. Thus, in this study, collagen was utilized as a natural compound and conduits were further modified by the addition of a conductive component, namely PANI, which has been reported to improve the efficiency of regeneration of injured nerves [15]. This combination of components allowed obtaining biocompatible conduits with sufficient wettability. Moreover, the mechanical testing of fabricated structures showed that the combination of components had good mechanical properties and elasticity, which is important in terms of avoiding post implantation chronic compression to regenerated nerve or collapsing of the conduit, which was previously reported for electrospun tubular scaffolds made of PLGA/PCL [24]. Our studies, both mechanical and in vivo, confirmed sufficient mechanical properties (resistance to collapsing). Moreover, we observed that ASCs cells grow inside of P(LLA-CL)-COL-PANI conduits, keeping their metabolic activity.

Biocompatibility studies showed that the addition of collagen to P(LLA-CL)-PANI fibrous meshes had a positive effect on the metabolic activity of cells seeded on fibrous meshes with higher viability of cells cultured on P(LLA-CL)-COL-PANI. Moreover, the improved neural differentiation of ASCs was observed for meshes containing protein, which is in accordance with findings reported by Carfi et al. and Gelain et al. [25,26].

Both qualitative and quantitative results of muscle tissue analyses indicated that the conduits reconstituted nerve continuity and prevented muscle atrophy, which occurred in control Group A, in a long, 6-month observation period. However, the addition of ASCs to the conduit slightly decreased the beneficial effect vs. the autograft group. It opposes other studies, where ASCs addition either to allografts or conduits was even more profitable for wet muscle mass ratio [27,28,29]. Yet, it was statistically significant that the differences between mean values of Groups B, C, and D remain much lower vs. similar studies. Recently, Nakada et al. reported the beneficial effect of enriching decellularized autografts with ASC, which enabled maintaining 60% of wet muscle mass ratio vs. the autografted group [28]. In our study, the conduit or its ASCs-enriched version achieved respectively 96% and 87% muscle mass ratio preservation with relation to the autografted group. Each intervention greatly decreased the extent of muscle fibrosis revealed by Masson’s Trichrome staining. The lowest level of fibrosis was observed in Group D; however, it was not statistically significant. Muscle mass retention was also mirrored by restoring the function of sciatic nerve during gait analysis. However, no significant differences were observed among Groups B, C, and D; conduits enriched with ASC achieved the highest values of index majority of time-points, which were followed by the autograft group and the unseeded conduit. As we discussed above, there is a small but significant difference in the mean wet mass weight of the muscle in group D vs. B, and it is insignificantly lower in D vs. C. Conversely, the addition of ASC, in Group D, produced the highest number of nerve fibers (Figure 10). Since we did not run specific analyses to answer that question, we can only drive assumptions. We presume that group size may have affected the results, since the difference is relatively small. 

ASC may have promoted the growth of sensory fibers more prominently than autograft or conduit alone; thus, we observed a higher fiber count but lower muscle mass. Further immunohistochemical and functional assays would be necessary to verify the hypothesis.

Neural tissue analysis revealed convincing superiority of P(LLA-CL)-COL-PANI conduits with ASCs versus autograft. Both Groups C and D improved nerve fiber density. It stays in line with other reports concerning the use of ASCs as an additive in nerve regeneration therapies [28,29]. The similarly beneficial effect of ASCs sheet enveloping a decellularized autograft was presented by Nakada et al.; however, their solution did not exceed the axonal density of the comparator (1.8 vs. 2.29 axons/µm^2^) as we observed in Group D vs. B in both stainings. We also observed a significantly improved G-ratio and circularity of nerve fibers in groups with an ASC-enriched conduit versus not seeded and autograft. A corresponding effect was reported by Saller et al. [30].

ASCs implantation in the PNI treatment protocol alters numerous parameters concerning nerve function. They increase nerve fiber or axon count as well as myelin sheath thickness, improve axon arrangement and nerve conduction velocity [21], improve sciatic function index (SFI) and muscle mass, and prevent atrophy and fibrosis [31,32]. Mainly, undifferentiated to ASCs (uASCs) are used in neuroregenerative studies, and such cells cannot form myelin itself (in contrast to ASCs differentiated to Schwann cells—ASCs-SC), but they produce neurite extensions [33]. Nonetheless, uASC could provide explicit support for nerve gap regeneration managed with a collagen I-based conduit, as demonstrated by Klein et al. [21]. However, Kappos et al. states that supreme regeneration conditions were provided by ASCs-SC compared to other ASC types (rat uASC, human superficial or deep uASC) [22]. Nakada along with other authors assumes that the beneficial effect of ASCs in conduits relies on the humoral effect of secreted neurotrophic factors [20,28,34]. As these studies show, undifferentiated ASCs are not viable after around 14 days after implantation [20] and do not provide myelinization, as they did in the case of processed allografts or conduits lacking living and interacting cells in their structure. Thus, a combined approach involving padding the conduit lumen with ASCs-SC and additional uASCs as a humoral support should be considered.

The study shows early results of implementing P(LLA-CL)-COL-PANI conduits in peripheral nerve regeneration. Up to date, there are no data on P(LLA-CL) conduits mixed with type I collagen and PANI. However, there are a few recent studies investigating P(LLA-CL) itself or similar derivates in PNI. They proved effective in improving nerve regeneration of the nerve gap [35,36] or nerve coaptation site wrapping [37]. 

PCL is also combined with other particles, which may facilitate nerve regeneration, such as laminin [38], graphene [39], carbon nanotubes [40], and small porcine intestine submucosa [41]. Thus, P(LLA-CL)-COL-PANI may serve as an effective platform for further enhancements and modifications yet exhibiting considerable proneuroregenerative effect. In the in vivo phase, we observed favorable features of tested conduits and their ASC-enriched derivate—in the context of muscle mass retention and regeneration of nerve fibers. We did not observe a massive influx of inflammatory cells or the presence of FBGCs-alike cells described by others [12].

The major limitation of the study relies on modest data concerning electroneurography and electromyography, which could reveal more data on functional outcomes of nerve regeneration. On the other hand, the muscle weight ratio and SFI results, which are also objective measurements of nerve regeneration, could support our findings. Another limitation is the lack of standardization in metabolic activity studies with plastic material as a control, although P(LLA-CL)-PANI used as a control instead of plastic was described in our previous studies (Appendix A and Appendix A). Moreover, Neurod 1 gene expression tests, which included plastic material as a positive control, supported the good biological properties of evaluated P(LLA-CL)-COL-PANI material. Another limitation of this study is the lack of evaluation of two peripheral nerve regeneration phenomena well illustrated by Fogli [42]: excessive sprouting [43] and backward projection [44] of regenerating nerve fibers. 

The P(LLA-CL)-COL-PANI-based scaffold examined in the study showed promising features as a novel alternative in the family of peripheral nerve conduits. Its efficiency in the management of the neural tissue gap is comparable with nerve autografts.

## 4. Materials and Methods

### 4.1. Materials 

Poly (L-lactic acid)-co-poly(ϵ-caprolactone) (P(LLA-CL)) with a ratio of 70:30 was purchased from Evonik (Germany). Atelocollagen (COL) was purchased from KOKEN (Tokyo, Japan). 1,1,1,3,3,3-Hexafluoro-2-propanol (HFP) was obtained from Fluorochem (Hadfield, UK). Polyaniline (PANI) and (1S)-(+)-10-camphorsulfonic acid (CSA) were purchased from Sigma-Aldrich (Saint Louis, MO, USA). Dulbecco’s Modified Eagle’s Medium (DMEM), fetal bovine serum (FBS), penicillin–streptomycin, Purelink RNA Mini kit, and phosphate buffer saline (PBS) were purchased from Thermofisher, USA. MSC Neurogenic Differentiation Medium was obtained from Promocell. Octenisept was purchased from Shűlke and Mayr (Norderstedt, Germany).

### 4.2. Fabrication of Electrospun Tubular Fibrous Scaffolds

P(LLA-CL) and collagen in a ratio of 85:15 were dissolved in HFP to form a 10% (*w*/*v*) solution and stirred overnight. To dope the polyaniline emeraldine base, PANI and CSA were mixed in a ratio of 50:50 in HFP solution and stirred overnight. Furthermore, both solutions were mixed in a ratio of 7 to 3 and stirred for an additional 12h. Then, the composite solution was electrospun [45] in the NANON-01 apparatus (MECC, Fukuoka, Japan) at a voltage of 15 kV using a needle to collect a working distance of 80 mm and a feed rate of 1.0 mL/h. A stainless-steel mandrel with a 2 mm diameter wrapped with a 0.1 mm copper wire rotating with a speed of 100 rpm was used as a collector. The 21 G needle—spinneret was moving with a speed of 100 mm/s, and the spinneret width was set to 160 mm. The electrospinning process was conducted for 70 min. Then, fibers collected onto mandrels in the form of tubes were dried for 48 h in vacuum drier (25 °C, 50 mbar). Furthermore, electrospun tubes were gently demolded from the mandrel by removing the copper wire and cut into segments of 20 mm for material properties evaluation and of 14 mm ready to use for in vitro experiments and in vivo as conduits. Prior to use, in all performed experiments, conduits were sterilized by Gamma radiation with a dose of 25 kGy.

### 4.3. Characterization of the Electrospun Scaffolds

The morphology of the obtained fibrous tubes was observed, after sputter coating with gold, using a scanning electron microscope (SEM, PhenomX, Eindhoven, The Netherlands) at an accelerating voltage of 10 kV. The diameters of obtained nanofibers were analyzed from the SEM images using Image Analysis Software (Madison, WI, USA). The average diameter of the fibers was determined by measuring the diameters of 50 randomly selected fibers. The thickness of the walls and the outer diameter of the tubes was measured using a micrometer screw in 5 different places of the tube.

The tensile properties of the electrospun nanofibrous tubes were evaluated using a tensile testing machine Instron 5943 (Instron, Norwood, MA, USA) at a crosshead speed of 5 mm/min under ambient conditions. The samples for mechanical tests were 20 mm long, and the end of each side of 5 mm was attached to the hydraulic clamps.

The hydrophilicty of the tubes was evaluated by the static contact angle measurement method using a contact angle goniometer (OCA 20, Dataphysics, Filderstadt, Germany). A volume of 2 µL of DI water was placed on the surface of the samples, and 3 s after the droplet touched the surface, the image was taken. Measurements were taken at nine different positions, and contact angles were calculated using SCA20 software.20.

### 4.4. Biocompatibility of the Scaffolds and In Vitro Studies

#### 4.4.1. Harvesting, Isolation, and Differentiation of ASCs

Ten male Lewis rats were sacrificed and euthanized for fat pad harvest. Fat was harvested in a sterile manner from 4 regions—inguinal, gonadal, pararenal, and nuchal (parascapular). Fat pads were collected in 1% PBS and immediately transferred on wet ice to the Department of Regenerative Medicine for further ASCs isolation. Each fat pad underwent isolation protocol with 0.075% Type I collagenase, which is consistent with the original protocol proposed by Zhu et al. and previous studies [46,47]. An isolated stromal vascular fraction (SVF) pellet was dissolved in 2 mL of DMEM. Cells underwent immediate and delayed (after 3–5 passage) cytometric analysis (CD29, CD11b, CD90, CD45 and CD34). SVF was also subjected to a differentiation test with alizarin red, Masson’s Trichrome, and oil red staining. Finally, MTS and doubling time tests were applied to compare the survival rate of fat harvested from different regions. Ultimately, there were no statistically significant differences in the survival rate of cells and contents in ASCs in any region. Thus, cell culture after the third to fifth passage from all regions was used to prepare ASCs stocks for conduits enrichment in further steps.

#### 4.4.2. Metabolic Activity of ASCs Cultured on P(LLA-CL)-COL-PANI and P(LLA-CL)/PANI Meshes

P(LLA-CL)-COL-PANI and P(LLA-CL)-PANI meshes were cut into 18 × 18 mm squares, placed into a 24-well plate, and pressed with sterile stainless-steel rings in order to prevent floating of the mesh fragments. Then, 500 µl of medium containing DMEM with 10% of FBS and 1% of penicillin–streptomycin was added into every well and incubated in 37 °C for 30 min. Afterwards, 7.6 × 10^3^ ASCs were seeded on each sample and incubated in 1 mL of DMEM with 10% of FBS and 1% of penicillin–streptomycin in 37 °C for 4 days. The metabolic activity of ASCs cultured on P(LLA-CL)-PANI meshes with or without collagen was assessed using Cell Titer 96 Aqueous One solution Cell Proliferation Assay (Promega, Madison, WI, USA). Meshes without the addition of collagen (P(LLA-CL)-PANI) were characterized in previous studies (Supplement) and used as a control for those with collagen (P(LLA-CL)-COL-PANI). Then, this medium was removed, and 160 μL of fresh medium and 40 μL of MTS reagent were added into each well containing cell-mesh construct. Cells were incubated in the dark for 2 h in 37 °C. Then, supernatant from each well was further transferred into a 96-well plate, and the absorbance of the obtained formazon dyes were measured using a spectrophotometric plate reader at a wavelength of λ = 490 nm using a Multiscan GO spectrophotometer (Thermofisher, Waltham, MA, USA).

#### 4.4.3. ASCs Neural Differentiation on P(LLA-CL)-COL-PANI and P(LLA-CL)/PANI Meshes

ASCs were cultured in growing medium on P(LLA-CL)-COL-PANI and P(LLA-CL)/PANI meshes for 7 days. Afterwards, the growing medium was removed, and 500 µL of differentiating medium (MSC Neurogenic Differentiation Medium) was added into each well. Cells were incubated for 4 and 7 days, and medium was changed every 2 days. In desired time points, the differentiation medium was withdrawn, and wells were washed twice with PBS. Then, 500 µL of 0.05% trypsin was added into each well and incubated for 5 min in 37 °C. Trypsin with detached cells was further transferred into a 15 mL tube containing 1 mL of FBS. Cells were centrifuged in 400× *g* for 7 min. Supernatant was taken out; then, cells were suspended in 1 mL of PBS and counted in Bürker chamber.

Neural differentiation was evaluated on the basis of Neurod1 gene expression, using the real-time PCR method. RNA was isolated from detached differentiated ASCs and placed in 15 mL Falcon tubes. Cells were centrifuged in 400× *g* for 7 min. Supernatant was withdrawn, and cells were transferred into Eppendorf tubes and centrifuged in 2000× *g* for another 5 min. A Purelink RNA Mini kit was used for RNA isolation according to the manufacture’s instructions. Then, supernatant was gently removed, and 300 µL of lysis buffer with 3 μL 2-mercaptoethanol was added and vortexed. Cell lysate was subsequently homogenized using a syringe with injection needle (size 18–21). Then, 70% ethanol was added into cell homogenate in a 1:1 ratio, and samples were further vortexed. The probe content was transferred into a tube with a column and centrifuged in 1200× *g* for 15 s, and supernatant was discharged. Then, 700 μL of washing buffer I was added into the column, and probes were centrifuged again. The probe with supernatant was discharged, and a column containing isolated RNA was transferred into a fresh tube, and 500 μL of washing buffer II was added. Next, probes were centrifuged in 12,000× *g* for 2 min, a further column was transferred into fresh probe, and 50 μL of rnase free water was added and incubated for 1 min in RT in order to wash out RNA from the column’s membrane. The quantity of isolated RNA was detected using an μDrop plate and spectrophotometer Multiscan GO (Thermofisher, Waltham, MA, USA). RNA was stored at −80 °C.

After RNA isolation, reverse transcription reaction was performed according to manufacturer’s instructions using a High-capacity cDNA Reverse Transctription Kit. The reaction mix, containing nuclease-free water, dNTP mix, RT buffer, RT random primers, Mulstiscribe Reverse Transcriptase, and isolated RNA were prepared on ice and gently vortexed in order to avoid air bubbles. Probes were placed in a thermocycler (MJ Research PTC-200 Gradient PeltierThermal Cycler, Marshall Scientific, Hampton, NH, USA), and reverse transcription reaction proceeded.

Neurod1 gene expression was detected using qPCR in a Lightcycler96 thermocycler (Roche, Switzerland), using Gadph and Actb as reference genes. Results were normalized using those genes’ expression. qPCR reaction was performed using TaqMan probes (Thermofisher, Waltham, MA, USA) and Universal MasterMix, NoAmpErase UNG (Thermofisher, Waltham, MA, USA) as well as cDNA, which was obtained during reverse transcription reaction. cDNA in concentration 100–150 ng/μL was used for qPCR reaction. The expression of Neurod1 genes in ASCs cultured on meshes containing or not containing collagen was compared. ASCs cultured on plastic plates with and without differentiation medium was used as a positive and negative control, respectively.

#### 4.4.4. Culture of ASCs on P(LLA-CL)-COL-PANI Tubular Scaffolds

Tubular fibrous scaffolds were sterilized using the radiation method with a dose of 25 kGy, which were cut into 14 mm fragments with a scalpel and placed into separate Petri dishes containing DMEM. After 1 h incubation in 37 °C with 5% CO_2_ and 95% humidity, electrospun tubes were transferred into 15 mL Falcon tubes containing DMEM with 10% FBS and incubated again in the same conditions. In order to seed cells into the scaffolds, 3 × 10^6^ ASCs were centrifuged in 400× *g* for 7 min, so the pellet volume was 50 μL. Scaffolds were held with sterile tweezers, and ASCs were injected into the tubes. Each tube with cells was put into separate Petri dish and incubated for 7 days in DMEM with 10% FBS in 37 °C with 5% CO_2_ and 95% humidity. After 7 days of culture, the medium was removed, and scaffolds with cells were transferred into another Petri dish. In order to evaluate the biocompatibility, cells were counted separately both within the scaffold and outside the scaffold in Petri dish. Then, 5 mL of 0.05% trypsin was added into a Petri dish and incubated for 5 min in 37 °C. Trypsin with detached cells was transferred into a 15 mL Falcon tube containing 1 mL of FBS, the Petri dish was washed twice with PBS, and PBS was also added into the tube. Cells were centrifuged for 10 min in 400 g, and cells’ number and viability were evaluated as described before. Scaffolds seeded with ASCs were transferred into different Petri dish, and 5 mL of 0.05% trypsin was added into each scaffold using an automatic pipette in order to wash out the cells. Petri dishes with scaffolds were incubated with trypsin for 5 min in 37 °C. Trypsin with detached cells was further transferred into the 15 mL Falcon tube containing 1 mL of FBS and scaffolds, and Petri dishes were washed twice with PBS and PBS was also added into the probes. Cells’ suspensions were centrifuged in 400× *g* for 5 min, supernatant was removed, and cells were counted as described before. The number and viability of the cells from the scaffolds and Petri dishes were compared. 

#### 4.4.5. Animal Study Design

Animal care and handling were carried out in accordance with the UK’s Animals (Scientific Procedures) Act 1986 and associated guidelines, the EU Directive 2010/63/EU for animal experiments, and they also complied with the ARRIVE (Animal Research: Reporting of In Vivo Experiments) guidelines. The experiments were approved by the Second Local Ethics Committee in Warsaw (Protocol no 72/2015). The number of animals required for the study was calculated with G*Power [48] with the following parameters: ANOVA, fixed effects omnibus, one-way, power—0.95, effect size 0.9, number of groups 4. All surgical procedures were performed using aseptic techniques. Inbred male Lewis rats aged 4–6 months (n = 28) were acclimatized to a 12 h light/dark cycle at 19 °C, with unlimited water and standard food. The rats weighed 366 to 486 g at the time of surgery.

Twenty-eight animals were randomly assigned into 4 even groups, each receiving a unilateral sciatic nerve transection and resection of 10 mm of its trunk and further: (A) control, 10 mm gap remained, (B) reconstructed with autograft, (C) reconstructed with P(LLA-CL)-COL-PANI conduit, (D) reconstructed with P(LLA-CL)-COL-PANI conduit and enriched with ASCs (Figure 12). Animals were anesthetized with Isoflurane. The randomly chosen limb was sterilized 3 times with Octenisept, and a 3 cm cut was performed along the dorsal side of a rat’s thigh. Gluteal muscles (GM) were divided to expose the sciatic nerve (SN). SN was cleared from the surrounding tissues and measured to obtain a continuous 10 mm trunk to be resected with at least a 5 mm margin from its divisions or branches. The previously marked fragment of SN was resected with a sharp blade. In rats assigned to Group A, SN was left without coaptation, with both stumps unsecured. Group B was reconstructed with the resected trunk of the sciatic nerve, which was rotated 180° and used as an autograft; finally, it was coaptated with 2 epineural sutures on each side (10/0 Prolene). In Group C, the P(LLA-CL)-COL-PANI conduit was placed between the proximal and distal nerve stump, which were further advanced about 2 mm into the lumen of the conduit and secured with 2 epineural sutures on each side (10/0 Prolene). In Group D, 3 × 10^6^ ASCs were seeded into a P(LLA-CL)-COL-PANI conduit and incubated for 3 days in 37 °C, 5% CO_2_, and 95% humidity in separate Petri dishes in DMEM with 10% FBS. Implantation was managed similarly to Group C. After conduit securing, an additional portion of 0.3 mL of cultured ASCs stock containing 1 × 10^6^ cells was injected with a 32G needle in the lumen of the conduit (in the space between both nerve stumps) (Figure 13). Afterward, gluteal muscles were secured with two 4/0 Vicryl sutures, and skin was closed with interrupted 4/0 Vicryl sutures. Rats were administered postoperatively with analgesics and remained under strict observation for 6 months. Operated limbs were secured for the prevention of autotomy or self-cannibalism. Scaffolds were sterilized with 25 kGy dose radiation prior to the implantation.

### 4.5. Sciatic Function Index (SFI) Analysis

The rats were examined on the manual walking track before intervention (day 0) and on 1, 2, 3, 4, 5, and 6 months after the surgery. Footprints obtained with stained footpads were digitalized and analyzed with ImageJ software. Print length, toe spread, and intermediate toe spread were calculated and used in formula reported by Bain et al. [49].

#### Samples’ Harvest and Histological Analyses

At 6 months of observation, rats were sacrificed by inhalation using isoflurane overdose (>4% *v*/*v*). Samples of conduits (or nerve autografts), ipsilateral, and contralateral gastrocnemius muscle were harvested. In Group A, discontinuation of SN was confirmed during the harvest. Sciatic nerves were cut out with a 5 mm margin on both proximal and distal sides of conduits; in the case of Group B, harvest was performed likewise. Additionally, SN and its surroundings were inspected for the presence of inflammation, scarring, fibrosis, or conduit dissolution. Freshly harvested muscles were weighted on a RADWAG AS220 electronic balance (Puszczykowo, Poland), and results were expressed as a ratio of ipsilateral to the contralateral muscle (denervated to innervated). Nerve samples were divided into 3 portions—(A) proximal to the conduit, (B) body of conduit, and (C) distal to the conduit. Furthermore, samples were immediately immersed in a 10% buffered formalin solution and transferred to the Department of Methodology (Medical University of Warsaw) for further processing. Samples underwent a standard protocol for sample embedding in paraffin. Subsequently, nerves were cut into 4 µm slices and muscles were cut into 10 µm slices. The samples of the muscles were stained with hematoxylin and eosin (HE) with standard protocol and covered with a coverslip. Images of muscle sections were captured under 400× magnification. Masson’s Trichrome staining was used to assess muscles’ fibrosis. Additionally, samples from 7 random, untreated GMs were added as healthy comparator. Deparaffinized sections were stained manually according to manufacturer guidelines (Sigma-Aldrich, HT15) and closed with a coverslip glass. Masson’s Trichrome stained slides were scanned to obtain whole slide images (WSI) at 40x magnification, Then, images were analyzed by a modified project designed by Sant’Anna et al. in CellProfiler [50]. Finally, the percentage of pixels containing collagen fibers in the section was calculated.

Nerve sections in the center of conduit/autograft were stained immunohistochemically with NF-200 primary antibody (N4142, Sigma-Aldrich, Munich, Germany). Staining was performed according to manufacturer protocol. Samples were counterstained with hematoxylin. Images of stained nerve sections were captured under 400× magnification. For more precise morphometrical evaluation, samples from each group were stained with Weigert-Pal (WP) method according to the protocol described by Kiernan et al. to stain myelin sheaths [51]. Quantitative assessment was performed for sections B. Nerve fibers were identified with the aid of a semi-automatic analysis tool—CellProfiler and adjusted protocol reported by Paskal et al. [52]. Nerve fiber density was described as the number of fibers in mm^2^ (axons/mm^2^) in NF-200 staining and WP staining. Axonal area (μm^2^), myelin thickness (μm), circularity (1—means a perfect circle, ratio of minor to major diameter of an object), and g ratio (ratio of axon radius to whole fiber radius) were assessed likewise in WP stained samples.

Any signs of neuroma were also evaluated by a blinded pathologist on nerve cross-sections according to following criteria: disorganized, with regenerating axon sprouts; erosion of the perineurium and epineurium; loss of funicular architecture; and the presence of intraneural fibrosis [53].

### 4.6. Statistical Analysis

Data from each study were collected in Excel (Microsoft, Redmond, WA, USA). Kolmogorov–Smirnov test and Shapiro–Wilk test were used to determine the distribution of data. T-test, univariate ANOVA, and post hoc Turkey’s tests were used for the analyses accordingly. GraphPad Prism 6 (GraphPad Prism, San Diego, CA, USA) was used for statistical analyses and plots preparation. The threshold of statistical significance is set at *p* ≤ 0.05.

## 5. Conclusions

P(LLA-CL)-COL-PANI conduits are biocompatible and provide effective support for axonal regeneration in peripheral nerve gap reconstruction in an animal model. P(LLA-CL)-COL-PANI conduits with ASCs enrichment produce a higher density of axons in reconstructed nerve but slightly decrease the wet muscle mass ratio vs. P(LLA-CL)-COL-PANI conduit alone. Further histological and functional studies of the conduits are indicated.

## Figures and Tables

**Figure 1 ijms-22-05588-f001:**
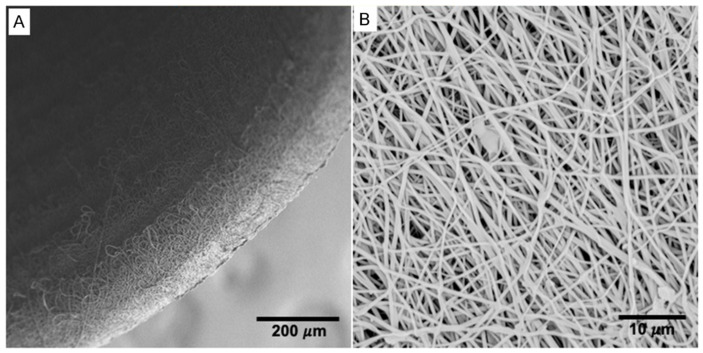
SEM images of electrospun P(LLA-CL)-COL-PANI conduits. (**A**) View into the inner part of a tubular scaffold; (**B**) Magnification on the structure of the scaffold.

**Figure 2 ijms-22-05588-f002:**
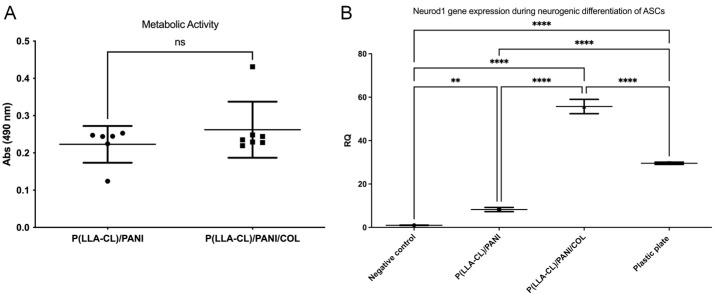
(**A**) Metabolic activity of ASCs cultured on P(LLA-CL)-PANI and P(LLA-CL)-COL-PANI electrospun fibrous meshes; (**B**) Neurod 1 gene expression during the neurogenic differentiation of ASCs on P(LLA-CL)-PANI, P(LLA-CL)-COL-PANI and plastic plate (positive control). Negative control—ASCs cultured on plastic without differentiation medium; Significant difference: ** *p* < 0.01; **** *p* < 0.0001.

**Figure 3 ijms-22-05588-f003:**
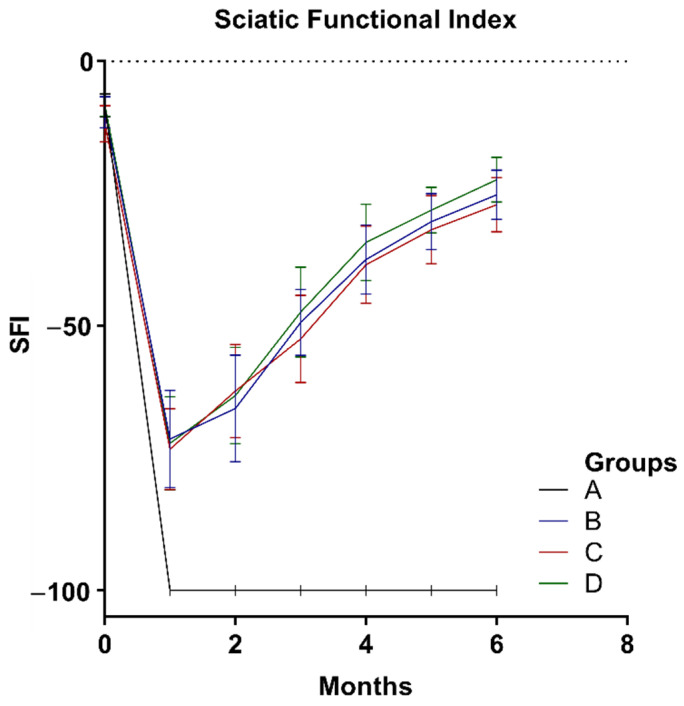
Graph represents sciatic function index (SFI) within 6 months observation in 1-month intervals obtained for examined groups: (A) nerve gap, (B) autograft, (C) P(LLA-CL)-COL-PANI conduit, (D) P(LLA-CL)-COL-PANI with ASCs. No statistically significant differences were noted among experimental groups.

**Figure 4 ijms-22-05588-f004:**
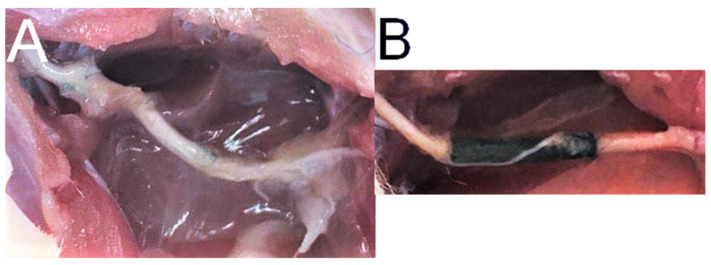
Six months after interventions, sciatic nerve were excised for further analyses. (**A**) Group B—sciatic nerve reconstructed with autograft after 6 months, (**B**) Group D—sciatic nerve reconstructed with conduits after 6 months.

**Figure 5 ijms-22-05588-f005:**
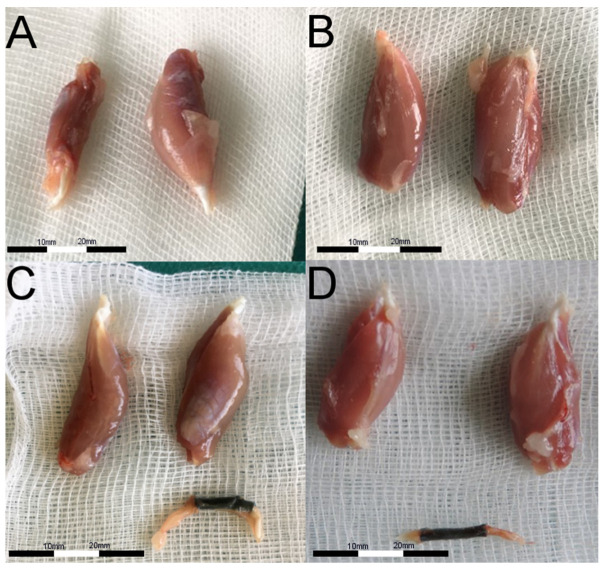
Postoperative images of gastrocnemius muscle excised after 6 months from interventions. On the left side—muscle from an operated limb, on the right side—muscle from the side after reconstructive procedure. (**A**) Control group—nerve gap remained, the muscle was innervated and decreased its size vs. unoperated limb on the right. (**B**) Autograft—muscle size is comparable. (**C**) P(LLA-CL)-COL-PANI conduit integrated into the sciatic nerve (bottom), which substantially preserved gastrocnemius size (left). (**D**) P(LLA-CL)-COL-PANI conduit was enriched with ASC cells, which prevented the muscle from atrophy (left vs. right).

**Figure 6 ijms-22-05588-f006:**
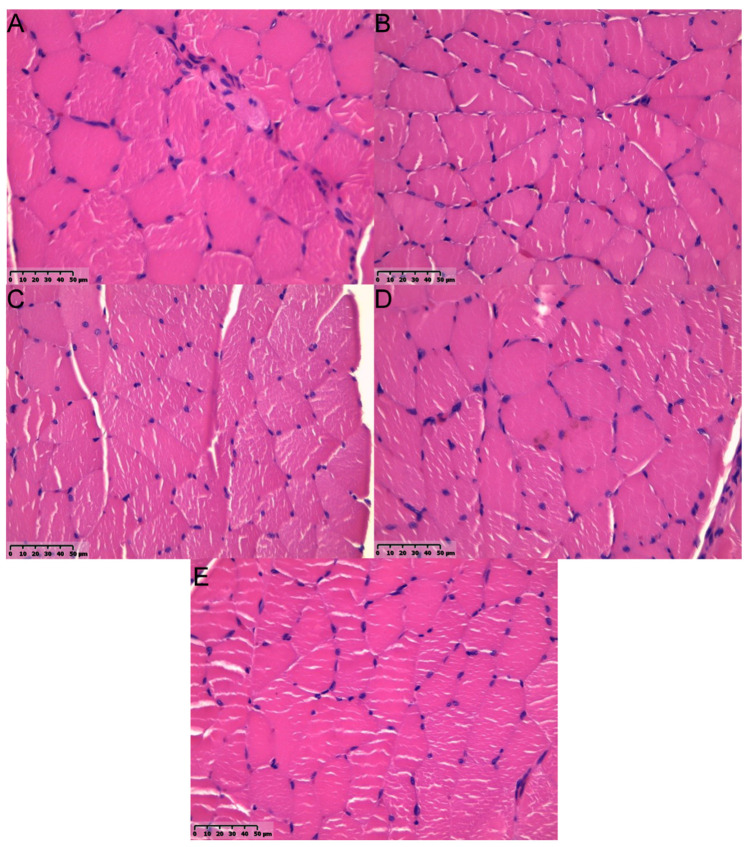
HE images of gastrocnemius muscle sections from the operated limb (**A**–**D**) and from random unoperated limb (**E**). (**A**) In the nerve gap group, the muscle fiber size is decreased; a higher number of nuclei was observed along with the increased prevalence of connective tissue between muscle fibers and inclusion bodies. (**B**) Autograft group—slightly increased number of inclusion bodies and increased number of nuclei and inclusion bodies. (**C**) P(LLA-CL)-COL-PANI conduit group—comparing to (**E**) slightly higher number of nuclei and connective tissue fibers were noted. (**D**) P(LLA-CL)-COL-PANI conduit with ASCs group—a higher number of inclusion bodies was noted. (**E**) Histologic image of a correct, unaffected muscle. Scale bar 50 µm.

**Figure 7 ijms-22-05588-f007:**
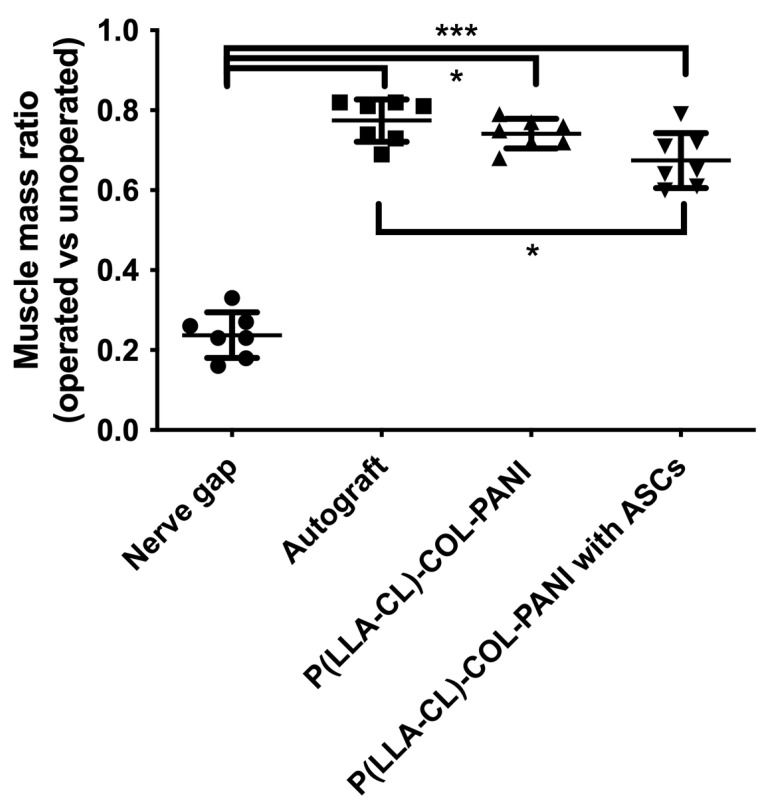
Diagram represents muscle mass ratio (operated vs. unoperated limb) among examined groups. Mean ± SD, * *p* < 0.05, *** *p* < 0.0001.

**Figure 8 ijms-22-05588-f008:**
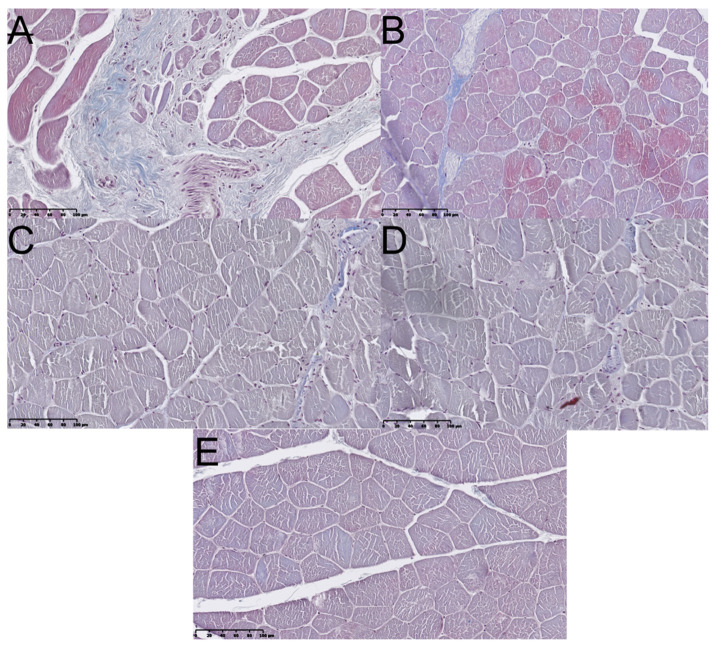
Masson’s Trichrom stained images of gastrocnemius muscle sections from the operated limb (**A**–**D**) and from random unoperated limb (**E**). (**A**) In the nerve gap group, an increased fibrotic tissue prevalence is present. (**B**) Autograft group—relatively low level of fibrosis within muscle bundles or their peripheral surrounding. (**C**) P(LLA-CL)-COL-PANI conduit group—image is comparable with group B. (**D**) P(LLA-CL)-COL-PANI conduit with ASCs group—image is comparable with group B and C. (**E**) Histologic image of a correct, unaffected muscle. Scale bar 100 µm.

**Figure 9 ijms-22-05588-f009:**
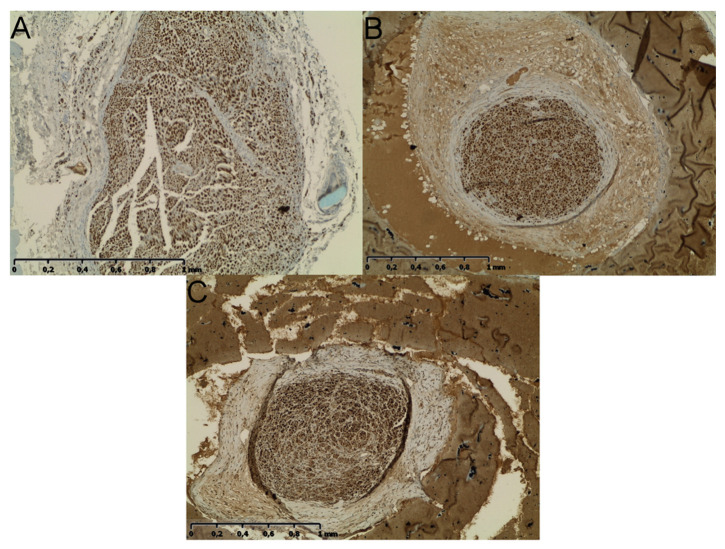
NF-200 immunohistochemistry images of nerve sections in the center of conduit/autograft. (**A**) Autograft group, gentle inclusion of fibrotic tissue with the graft was noted, a high number of well-developed nerve fibers were present with medium-sized myelin sheaths, regular morphology of perineurium. (**B**) P(LLA-CL)-COL-PANI conduit group—fibers are densely encapsulated within the lumen of the conduit, the perineurium is significantly developed and often exceeds 50% of the area of the conduit lumen, yet without signs of fibrosis, scarring, or compression within nerve fiber tissues. The outer layer consists of a partially dissolved conduit (**C**) P(LLA-CL)-COL-PANI conduit with the ASC group—the density of nerve fiber within the epineurium sheath is high, without scarring of fibrosis. The perineurium is thick, but to a lesser extent vs. conduit alone, it does not fully fill the lumen of a conduit. The outer layer consists of a partially dissolved conduit. Scale bar 1 mm.

**Figure 10 ijms-22-05588-f010:**
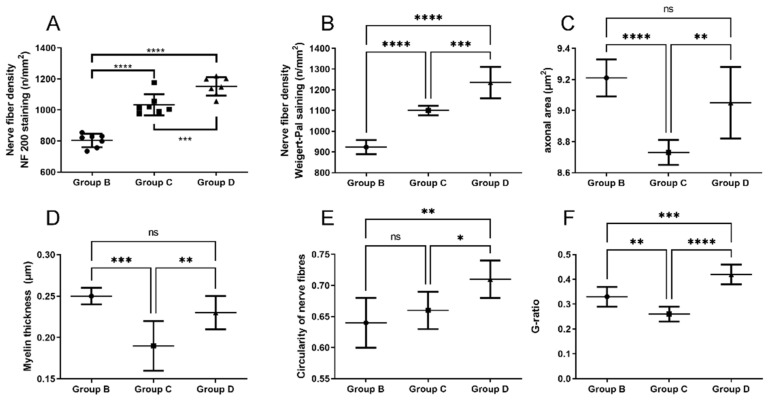
Nerve fibers morphometry (Group B—nerve autograft, Group C—P(LLA-CL)-COL-PANI conduit, Group D—P(LLA-CL)-COL-PANI with ASCs): (**A**) Diagram represents nerve fiber density calculated from NF200 stained images. (**B**–**F**) Diagrams represent nerve fiber density (**B**), axonal area (**C**), myelin thickness (**D**), nerve fiber circularity (**E**), G-ratio (**F**) calculated from Weigert-Pal stained images. Mean ± SD, * *p* < 0.05, ** *p* < 0.01 *** *p* < 0.001, **** *p* < 0.0001.

**Figure 11 ijms-22-05588-f011:**
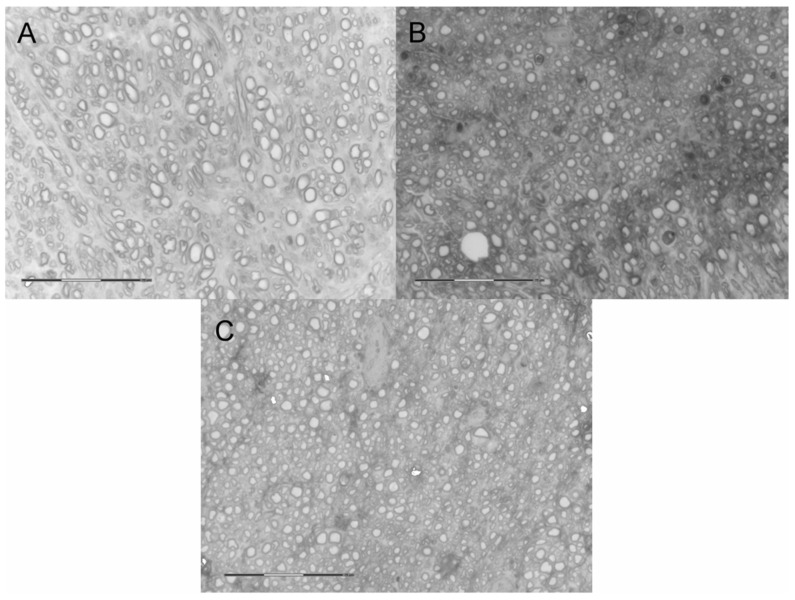
Weigert-Pal staining images in of nerve sections (800×, grayscale). (**A**) Autograft group; comparing to other groups, there is a lower density of fibers, which are relatively larger and have a thicker myelin sheath. (**B**) P(LLA-CL)-COL-PANI conduit group—fibers are more densely arranged, and there are a larger number of artifacts and inclusions. (**C**) P(LLA-CL)-COL-PANI conduit with ASC group—density of nerve fibers is comparable with Group B, there are less inclusions and scar tissue.

**Figure 12 ijms-22-05588-f012:**
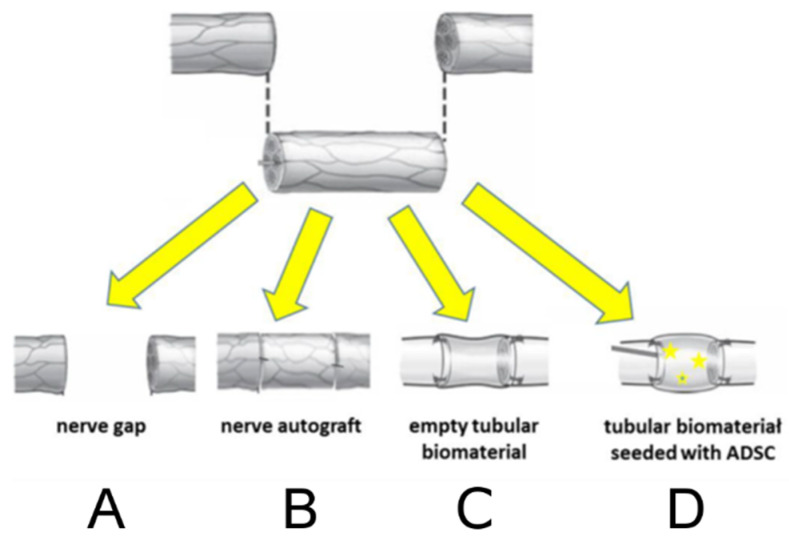
Diagram represents four interventions (groups) to the sciatic nerve. After sciatic nerve transection and excision of 10 mm trunk: (**A**) Nerve gap remained; (**B**) Excised trunk was rotated 180° and used as an autograft; (**C**) P(LLA-CL)-COL-PANI conduit was inserted in the gap; (**D**) P(LLA-CL)-COL-PANI enriched with ASCs was inserted in the gap.

**Figure 13 ijms-22-05588-f013:**
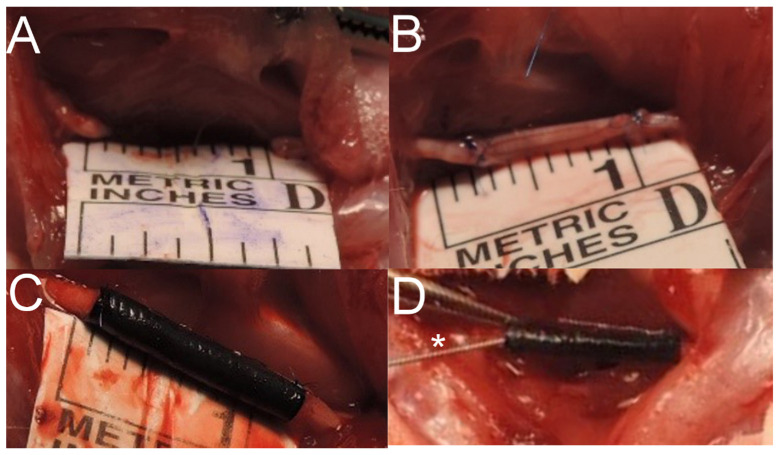
Intraoperative photograph of each experimental group. (**A**) After excision of 10 mm trunk of the sciatic nerve 10 mm gap remained, and the length was confirmed with a millimeter ruler; (**B**) Excised trunk was rotated and sutured to the nerve stumps with two 10/0 Prolene epineural sutures; (**C**) P(LLA-CL)-COL-PANI conduit filled the gap, nerve stumps were placed 2 mm into the lumen of the conduit and secured with two 10/Prolene epineural sutures; (**D**) In Group D, after the ASCs were seeded in the conduit, an implantation injection of 1 × 10^6^ ASCs was made into the lumen of the conduit (*—needle for ASC injection).

**Table 1 ijms-22-05588-t001:** Parameters of P(LLA-CL)-COL-PANI conduits.

Parameter	Value	Units
Wall thickness	272 ± 12	µm
External diameter of a conduit	2.04 ± 0.04	mm
Mean fiber thickness	460 ± 143	nm
Length of a conduit	15	mm
Tensile strength	8.53 ± 1.43	MPa
Elongation to break	332.25 ± 72.11	%
Contact angle	81.23 ± 2.14	°

**Table 2 ijms-22-05588-t002:** Automated analysis of fibrosis of muscle samples.

Group	Fibrosis
	% of Positively Stained Pixels (Mean ± SD)
A	65.34 ± 12.3
B	13.61 ± 3.42
C	15.38 ± 2.81
D	12.11 ± 2.76
Healthy muscle (*n* = 7)	5.11 ± 2.23

## Data Availability

Not applicable.

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
