# Peer review of "Bioactive Nanofiber-Based Conduits in a Peripheral Nerve Gap Management—An Animal Model Study"

_ijms, 2021, doi:10.3390/ijms22115588_

Round 1
Reviewer 1 Report
Introduction: This section seems quiet wordy. I would not consider conduits as a 'novel alternative' (line 57) as conduits have been investigated in-vitro and in-vivo for more than one decade.
Results: The images of Fig. 4 are underexposed. They should be edited or maybe replaced.
Discussion: There are a few language mistakes such as in lines 246, 247, 259. Further, there are several studies in this field that should be discussed in this context: PMID: 20851070, PMID: 27348645, PMID: 25942148
Methods: It is not clear how the authors determined the sample size. Was there a power calcualtion prior to the trial? Was the pathologist blinded? When was the rat euthanized that suffered the infection?
Reviewer 2 Report
The authors present a preclinical study about a scaffold made of poly (L-lactic acid)-co-poly(caprolactone), collagen (COL), polyaniline (PANI) and enriched with adipose-derived stem cells (ASCs), as a nerve conduit in a rat model of peripheral nerve injury. Such conduit showed promising results by decreasing muscle atrophy in a murine model
Due to the extremely limited therapeutic possibility in nerve injuries, when a graft is necessary, every new addition to the surgical armamentarium is welcome.
I have only minor issues. I.e, lines 57-58: "Conduits emerged as a novel alternative to these solutions
and struggle their way to the algorithms in the clinical approach".
I suggest adding some previous experiences with the role of collagen tube filled with autologous skin-derived stem cells for repairing the poly-injured motor and sensory nerves of the upper arms of a patient (see Grimoldi N et al, Stem cell salvage of injured peripheral nerve. Cell Transplant. 2015;24(2):213-22. doi: 10.3727/096368913X675700)
